# Prediction of future aging-related slow gait and its determinants with deep learning and logistic regression

Alison Deatsch[1]*, Michael McKenna[2], Jonathan Palumbo[2], Qu Tian[3],
Eleanor Simonsick[3], Luigi Ferrucci[3], Robert Jeraj[1,4], Richard G. Spencer[2]

1 Department of Medical Physics, University of Wisconsin-Madison, Madison, Wisconsin, United States of America, 2 Magnetic Resonance Imaging and Spectroscopy Section, Laboratory of Clinical Investigation, National Institute on Aging, National Institutes of Health, Baltimore, Maryland, United States of America, 3 Longitudinal Studies Section, Translational Gerontology Branch, National Institute on Aging, National Institutes of Health, Baltimore, Maryland, United States of America, 4 Faculty of Mathematics and Physics, University of Ljubljana, Ljubljana, Slovenia

☉ These authors contributed equally to this work.
* deatsch@wisc.edu

## Abstract

### Background

Identification of accelerated aging and its biomarkers can lead to more timely therapeutic interventions and decision-making. Therefore, we sought to predict aging-related slow gait, a known predictor of accelerated aging, and its determinants.

### Methods

We applied a deep learning neural network (NN) and compared it to conventional logistic regression (LR) analysis. We incorporated 1,363 participants from the Baltimore Longitudinal Study of Aging to predict current and future slow gait at 6-year and 10-year follow-up using two clinically-relevant cut-points.

### Results

Our NN achieved a maximum sensitivity (specificity) of 81.2% (87.9%), for a 10-year prediction with 0.8 m/s cut-point. We demonstrated the necessity of class balancing and found the NN to perform comparably to or in some cases, better than, LR which achieved a maximum sensitivity and specificity of 84.5% and 86.3%, respectively. Sobol index analysis identified the strongest determinants to be age, BMI, sleep, and grip strength.

### Conclusions

The novel use of a NN for this purpose, and successful benchmarking against conventional techniques, justifies further exploration and expansion of this model.

**Data availability statement:** The data that support the findings of this study are available from the BLSA (NIA) at this link (https://www.blsa.nih.gov/), but restrictions apply to the availability of these data. These restrictions are due to ethical considerations for informed consent to share human data. Consent to share data publicly was not included in the BLSA consent until recently, therefore most of the measurements used in these analyses are restricted by the IRB to "permissioned" access only. Analysis plans must be approved prior to gaining access to the data and a BLSA investigator must be included in the research team. Follow the instructions here (https://blsa.nih.gov/blsa-data-use) to register an account and request access to the data.

**Funding:** The author(s) received no specific funding for this work.

**Competing interests:** The authors have declared that no competing interests exist.

**Abbreviations:** NN, Neural network; LR, Logistic regression; DL, Deep learning; CNN, Convolutional neural networks; BLSA, Baltimore Longitudinal Study on Aging; BMI, Body mass index; AUPRC, Area under the precision-recall curve; MMSE, Mini Mental State Exam; ELU, Exponential Linear Unit; SCCE, Sparse categorical cross entropy; SAG, Stochastic average gradient; L-BFGS, Limited-memory Broyden-Fletcher-Goldfarb-Shanno; RUS, Random Undersampling; SMOTE, Synthetic minority oversampling technique; SMOTE-ENN, SMOTE with edited nearest neighbors undersampling; SALib, Sensitivity analysis library

# 1 Introduction

Understanding the hallmarks and predictors of healthy aging is essential to the goal of increasing population health span [1]. In particular, the development of tools to identify healthy versus accelerated aging trajectories and their early and longitudinal biomarkers can advance opportunities for more effective interventions and better-informed clinical decision-making.

One important indicator of biological aging is gait speed. Every decrement of 0.1 m/s in gait speed is associated with a 12% higher mortality in older adults [2] Among cohorts of older adults, gait speed has also been shown to be associated with survival, to reflect health and functional status, and to be a clinical indicator of well-being [3–10]. In fact, gait speed has been recommended as a primary endpoint for clinical trials due to the strong association between physical performance measures and brain age [11–15]. Slowing gait speed has also been shown to precede cognitive decline and to be associated with Alzheimer's pathology, making it a useful marker for neurodegenerative risk prediction [16]. Gait speed is a metric of particular relevance for those with long-term, chronic conditions such as COPD and heart failure as a significant predictor of functional status, future well-being, and development of diabetes [17–19]. Indeed, one of the chief advantages of gait speed as a health metric is the fact that it is impacted by the integrity of a wide range of organ systems, including neurological status (both sensory and motor), cardiovascular health, orthopedic status, and pulmonary function. However, despite the growing importance of gait speed as a marker for overall health, identification of risk for future decreased gait speed remains incompletely understood.

Since there is no single comprehensive predictor of biological aging, aggregate measures are needed that incorporate complementary clinical biomarkers [20]. Ideally, these would incorporate predictors encompassing functional and physiological domains essential for the study of aging and gait speed. Many studies have been performed correlating a current gait speed measurement with healthy aging metrics, [21–27] and baseline gait speed is often used as a predictor of other aging outcomes [4,28–33]. However, in spite of its importance, there are few studies attempting to predict future slow gait.

Many predictors have been considered for their predictive power to identify current and future gait speed. A number of investigators have considered similar sets of modifiable risk factors, medical conditions, and clinical data including age, sex, comorbidities, strength, sleep, and alcohol consumption [34–36]. Other studies have added more complex factors such as cognition, [37,38] inflammatory markers, [39] and brain volumes [35]. While many variables have been well-established to correlate with gait speed, the quantitative power of these to predict future gait speed decline is much less established.

Several attempts have been made using statistical models to predict gait speed changes from a narrow set of potential predictors, [34,40] with the most commonly-applied analytic techniques being linear and logistic regression. These approaches, however, are restricted to the exploration of only linear relationships, while the

complexity of human biochemistry and physiology suggests that their performance may be surpassed by that of models incorporating nonlinear effects and interactions. Indeed, there is a great deal of evidence for nonlinearity in human biological systems (e.g., circadian rhythms, calcium signaling, heart rate variability, disease dynamics) [41,42]. In the case of interest, several studies have previously found nonlinear relationships between gait speed and BMI, [43] age, [44] physical activity, [45] leg strength, [46] and falls [47]. Thus, it is essential that models predicting gait speed decline can encompass potentially nonlinear relationships.

Deep learning (DL) neural networks (NNs) permit the investigation of nonlinear model-free relationships between clinical variables and outcomes and have demonstrated major successes throughout the biomedical sciences. When employed in studies of aging, NNs and other machine learning methods have demonstrated high performance in defining and predicting functional and cognitive outcomes, with multiple contributions highlighting the importance of capturing nonlinear relationships [32,33,48]. For example, Lin et al. found that a NN outperformed logistic regression for predicting mortality in elderly patients with hip fracture [49]. However, despite this demonstrated early success, application of NNs to studies of aging, and to future gait speed prediction in particular, has been underexplored [50].

Indeed, there remains a gap in the literature regarding the use of NNs for prediction of gait speed from clinical variables. This is likely due in large part to the need for a substantial training dataset incorporating the relevant predictor and outcome variables for model development. The required number of subjects depends on the complexity of the analysis, the quality of the data, and the algorithm under consideration. For example, the number of subjects in a selection of comparable studies were 108, 239, 746, and 1901 [21,32,33,48]. Clearly, dataset needs for DL tasks can vary widely, but are generally larger than the required size for LR or basic statistical analysis.

In addition to providing a natural means of developing implicit nonlinear models, NNs exhibit much greater flexibility in the selection of input variables as compared to logistic and linear regression. Convolutional neural networks (CNN) also offer the opportunity to use raw image data without feature extraction, while separate NN structures can make full use of longitudinal data. [51]. Thus, our motivation for use of a NN included establishing a platform to be further expanded in subsequent work.

The development of models which can accurately predict current and future gait speed decline would be of great clinical use in several contexts. For example, these findings may inform physicians of potentially modifiable risks that could ameliorate potential loss of normative gait function in their patients. Similarly, physiotherapists may have the opportunity to design therapeutic protocols based on patient status.

The purpose of this work, therefore, was two-fold. First, we sought to predict aging-related slow gait and its determinants across various timeframes from a basic set of health measures. We employed a NN classification model to capture nonlinear complexity and compared its performance to a conventional logistic regression to demonstrate its viability. This defines our second goal, which was to develop a flexible NN architecture suitable for modification and implementation in further related studies.

## 2 Methods

### 2.1 Data

Our studies were performed using data from the Baltimore Longitudinal Study on Aging (BLSA) [52,53]. The BLSA is America's longest running scientific study of human aging, with data collection beginning in 1958. This provided sufficient data for training and permitted us to perform our studies on a population exhibiting, overall, normative aging. Our analysis focused on two clinically relevant gait speed cut-points (0.8 m/s and 1.0 m/s) [4,21,54–57]. For this pilot study, we chose input variables based on other studies evaluating current and future gait speed [21,23,26,27,34–38,58]. We first explored the identification of current slow gait speed and its determinants, but with this rich dataset, we were also able to investigate multiple prediction timeframes to identify trends in healthy aging and its determinants across time.

**2.1.1 BLSA dataset.** The BLSA was founded in 1958 and contains data on over 3,200 participants. The approximately 1,300 participants still active in the study return every 1–4 years to receive comprehensive health, cognitive, and functional evaluations. A freeze of the BLSA database was accessed August 17, 2021, and the authors performing the analysis had no access to information that could identify individual participants at any time. At the time of this study, the BLSA database consisted of 3,821 gait speed measurements from 1,363 unique subjects. Measurements were performed at intervals depending on subject age, with intervals between visits decreasing with increasing age (<60 = 4 yr intervals, 60–79 = 2 yr intervals, 80+ = 1 yr intervals). The median number of visits per subject was two, with a range of up to 12. This dataset is uniquely suited to our purpose due to the large number of subjects, the high quantity of metrics recorded, the regularity of the testing intervals, and the longitudinal nature of the study.

Gait speed measurements were obtained from timed walks performed according to the Short Physical Performance Battery (SPPB) protocol [59]. The SPPB, including its gait measurement component, is a standardized, widely used tool for measuring physical performance [60]. Participants were timed as they walked unassisted at a normal pace on a 6-meter course. Walking times were converted to gait speed (m/s) and the average of two trials was calculated. These timed walks have high test-retest reliability with ICC values 0.87–0.97 and uncertainty of 0.06–0.11 m/s, depending on the measured population [61–64].

Gait speed outcomes were evaluated using two different cut-points that have been shown to be clinically relevant for normatively aging population represented by the BLSA: (1) 0.8 m/s which has been proposed as a marker for severe mobility disability with a strong correlation to mortality [55–57] and (2) 1.0 m/s, the speed below which the risk of mortality doubles and at which subjects are deemed at high risk for adverse health-related outcomes [4,21,54]. Binary classification was chosen due to its natural clinical interpretation, similar to a host of other clinical outcome measures. In addition, both LR and NN architecture lend themselves naturally to such classification analyses. Indeed, a classification study is a natural design for a NN and allows us to compare the NN to the well-defined, gold standard LR model. In addition, the use of these cut-points provides clinical relevance as they are metrics for health status categorization and treatment guidance.

The number of gait speed measurements in the BLSA database for each cut-point is shown in Table 1. Any of these classifications (columns in the tables) can be used as an output for the DL model. The "Current" timeframe denotes the classification of gait speed at the time of the measurement, where a "Slow" measurement captures any gait speed that is at or below the cut-point value at that time. In the "Future" timeframes, "slow" captures any gait measurement above the cut-point value corresponding to a subject who will develop a slow gait, that is, fall below the cut-point value, within the time frame indicated. Subjects included in the Future timeframes were restricted to only those with at least as many years of follow-up as specified by the timeframe. In this way, subjects who will develop slow gait in the future or who developed this outside of the study were not incorrectly included in the "Normative" class. Other timeframes were also explored but proved to have too few subjects in the "Slow" class to achieve reliable results.

**Table 1. Counts of gait speed measurements and key demographics for each dataset from the BLSA database.**

| Cut-point | Dataset | Number of Slow Measurements (unique participants) | Number of Normative Measurements (unique participants) | Total Number of Gait Speed Measurements | Mean Age | % Male | % Slow |
|---|---|---|---|---|---|---|---|
| 0.8 m/s | Current | 276 (238) | 3545 (3066) | 3821 | 71.4 ± 13.3 | 49.4 | 7.22 |
| | Future: 6 Years | 181 (98) | 830 (504) | 1011 | 69.3 ± 11.6 | 48.6 | 17.9 |
| | Future: 10 Years | 245 (106) | 247 (221) | 492 | 71.8 ± 12.0 | 47.8 | 49.8 |
| 1.0 m/s | Current | 910 (789) | 2911 (2538) | 3821 | 71.4 ± 13.3 | 49.4 | 23.8 |
| | Future: 6 Years | 274 (182) | 662 (424) | 936 | 68.5 ± 11.2 | 49.6 | 29.3 |
| | Future: 10 Years | 392 (209) | 186 (167) | 578 | 70.9 ± 10.9 | 49.0 | 67.8 |

The number of unique participants from which the gait speed measurements arise is listed in parentheses.

In order to compare our results to previous models, we selected a subset of 15 markers encompassing demographic, lifestyle, and medical history variables previously identified as potentially correlated to gait speed decline either at current [21,23,26,27,34,38,58] or future [34–37] timepoints. Inclusion decisions were made based on several factors: (1) the amount of missing data for a particular variable across multiple time frames and subjects, (2) the typical availability of the variable in clinical settings, and (3) the interpretability of the variable. We preferred to incorporate fewer, well-selected variables, rather than more of the available variables, in order to increase the overall interpretability of the result and to prevent the difficulties arising from multiple dependencies. We considered age, sex, hypertension, diabetes, stroke, heart attack, osteoarthritis, cognitive impairment, depression, sleep quality, exercise, grip strength, pain, alcohol consumption, and BMI (Table 2). Categorical variables are noted in the table with (y/n) and were input as 0 or 1 to the model (or a scale of 1–5 in the case of sleep quality). Continuous variables (age, cognitive impairment, grip strength, and BMI) were input as exact values, as recorded in the BLSA data. Missing data points were replaced by the median value of that input for all participants, as is common practice in machine learning. Median was chosen in preference to the mean for this imputation to avoid undue sensitivity to outliers. The percentage of missing values was less than 2% for all variables except MMSE, which was missing for 15% of the gait speed measurements.

**2.1.2 Consideration of dataset bias.** We evaluated several points of potential bias in the BLSA data used for this study. Firstly, highly correlated input variables can pose a problem for logistic regression models, making it harder to interpret coefficients and identify significant independent variables. While this multicollinearity is less of a confounding factor for NNs, it can still slow convergence and affect sensitivity analysis. We checked the correlation of our input variables by calculating the Pearson correlation coefficient between each pair. The resulting coefficients for each pair are shown in S1 Fig in the Supporting Information. Values $< -0.5$ and $>0.5$ are generally considered to indicate notable correlations. In our case, we see this only for age and grip strength, with $r = 0.54$. We conclude that there was minimal multicollinearity in our input data. In fact, we elected to incorporate only the right-hand grip strength in the analysis due to high collinearity with the left-hand grip strength ($r = 0.74$). The use of dominant hand grip strength was also considered but would have further restricted dataset size since this value was not reported in all participants.

For the use of longitudinal data, the potential for bias due to selective dropout, or non-random loss of participants from a study, must be considered [65]. We compared the age, MMSE score, and gender proportion between the cohort of

**Table 2. Definitions of each input variable from the BLSA database.**

| Input Variable | Definition/ Survey Question |
| --- | --- |
| Age | |
| Sex | Gender (binary) |
| Hypertension | Has a doctor ever told you that you are hypertensive? (y/n) |
| Diabetes | Has a doctor ever told you that you have diabetes? (y/n) |
| Stroke | Has a doctor ever told you that you had a stroke, mini stroke, or slight stroke? (y/n) |
| Heart Attack | Has a doctor ever told you that you had a heart attack? (y/n) |
| Osteoarthritis | Has a doctor ever told you that you have osteoarthritis? (y/n) |
| Cognitive Impairment | Mini Mental State Exam (MMSE) score (0–30) |
| Depression | Has a doctor ever told you that you have depression? (y/n) |
| Sleep Quality | Sleep quality rating past month (Scale: 1–5) |
| Exercise | Have you performed vigorous exercise in past 2 weeks? (y/n) |
| Grip Strength | Hand grip muscles right (kg) |
| Pain | Have you had any/frequent pain in past year (overall)? (y/n) |
| Alcohol Consumption | Have you consumed alcohol in the past 12 months? (y/n) |
| BMI | Body Mass Index |

patients included in each dataset and those dropped due to lack of follow-up (see S1 Table in the Supporting Information). The average of each metric for the included cohort was compared to that of the dropped cohort. Potential bias was evaluated by two-sample t-tests as well as effect size calculations for age and MMSE score and by a two-proportion z-test for the percent males. Any significant differences from the t-test or z-test are denoted by the bold text in the table. The small effect sizes, all less than 0.22, as well as the fact that even statistically significant differences were well within one standard deviation of each other, indicate minimal bias due to selective dropout.

We also considered the possibility of dataset drift, that is, whether the average gait speed of subjects has changed over the many years of BLSA data collection. To evaluate this, we calculated the mean gait speed of subjects grouped by the year in which the measurement was taken. S2 Fig in the Supporting Information shows the lack of a clear trend, indicating absence of substantial drift. We also considered dataset drift with respect to year of birth, as shown in the Supporting Information in S3a Fig; such drift could, for example, result from a secular improvement in overall population health. To separate this from the expected trend with respect to age, we also plotted gait speed versus participant age in S3b Fig. A linear regression analysis resulted in β coefficients of 0.0083 and 0.0082, respectively. Based on the near equality of these coefficients and the shape of the plots, we conclude that the effect of dataset drift on our study due to birth year was also not significant.

Lastly, we considered the possibility of bias from the uneven distribution of participant ages at each timepoint of measurement. We found that there were far more measurements in the middle age range of 60–80 than at the extremes within the dataset. To explore the potential of bias due to this overrepresentation, we examined the pattern of false predictions with respect to age within our current timeframe models. We found that the shape of the age histogram closely matched the shape of the false prediction histogram for both cut-points, suggesting that prediction errors were occurring in proportion to the number of subjects across the dataset, that is, accuracy was relatively independent of age. We conclude that training set size was sufficient across the participant ages and there was consequently minimal bias attributable to sample age distribution.

**2.1.3 Ethics approval and consent to participate.** Written informed consent was obtained from all participants and the Institutional Review Board of the Intramural Research Program approved the study protocol.

## 2.2 Models

Two types of binary classification models were developed to predict slow or normative future gait speed. Separate models were created to predict a participant's current gait speed class. In all models, stratified 10-fold cross validation was performed. Each model type was trained and tested separately with unbalanced data and balanced data for the various timepoints and the two cut-points described in Section 2.1. Throughout this paper, we refer to each separately trained model with each dataset as an individual classifier. To be clear, sensitivity in our context reflects the ability to correctly classify participants with gait speed below the specified cut-point (also referred to as "slow walkers"), while specificity reflects the ability to correctly classify those with gait speed above the cut-point (also referred to as "normative walkers").

Our dataset is highly unbalanced, as is often the case for clinical data. In particular, the slow walkers are much less well-represented as compared to normative walkers. As a result, accuracy (proportion of correctly classified subjects) is a poor metric for performance evaluation; high accuracy may be attained even with poor sensitivity for identification of slow walkers; this is a well-known issue in clinical classification settings. In contrast, the Youden index (Sensitivity + Specificity − 1) provides a single summary statistic that captures both sensitivity and specificity. We therefore used this metric to optimize model hyperparameters and to compare performance across models. In addition, model performance was also assessed by sensitivity and specificity separately, precision, and the area under the precision-recall curve (AUPRC).

**2.2.1 Neural network methodology.** We implemented feed-forward NNs with the Keras library using TensorFlow as backend, using Python version 3.6. We chose a relatively simple NN architecture in order to incorporate nonlinear relationships without overcomplicating the model. An architecture with a limited number of nodes was chosen to prevent

overfitting since the input data (a single column, short length vector) was relatively non-complex compared to many DL tasks. Overfitting was mitigated by careful hyperparameter tuning and 10-fold cross validation as detailed below.

A schematic overview of our model is shown in Fig 1. Generally, the model used operational blocks as indicated by the inset of Fig 1; these consisted of a fully connected layer followed by batch normalization, an exponential linear unit (ELU) activation function, and dropout. Batch normalization was used prior to the activation function to standardize inputs and facilitate training. Dropout regularization was implemented to reduce overfitting and improve generalizability. Four such sequential FC blocks were followed by another fully connected layer and then by a final softmax binary output layer. Further details on model optimization through loss function testing and hyperparameter tuning can be found in the Supporting Information.

**2.2.2 Logistic regression methodology.** Logistic regression was also implemented using the scikit-learn module in Python for benchmarking our DL results. We compared performance using several solvers: Newton's method, liblinear, stochastic average gradient (SAG), and limited-memory Broyden-Fletcher-Goldfarb-Shanno (L-BFGS). We found no significant difference between these, and so selected the default L-BFGS solver with L2-regularization. Inputs to this model were the 15 physiological markers used for the NN method, and the target variable was again a binary slow/normative label assigned based on either the 0.8 or the 1.0 m/s cut-point. Stratified 10-fold cross validation was again used, with results compiled from the average of the 10 folds. Logistic regression was performed both with and without the use of terms representing first order interactions with age.

## 2.3 Class balancing techniques

As in many clinical studies, the BLSA dataset exhibited substantial class imbalance, with many fewer slow than normative walkers. Models trained with unbalanced datasets often serve as poor predictors of the minority class; [66] therefore, we explored the efficacy of various class balancing techniques. We compared the effect on model performance from one undersampling technique, Random Undersampling (RUS), and two oversampling techniques, synthetic minority oversampling technique (SMOTE) and SMOTE with edited nearest neighbors undersampling (SMOTE-ENN).

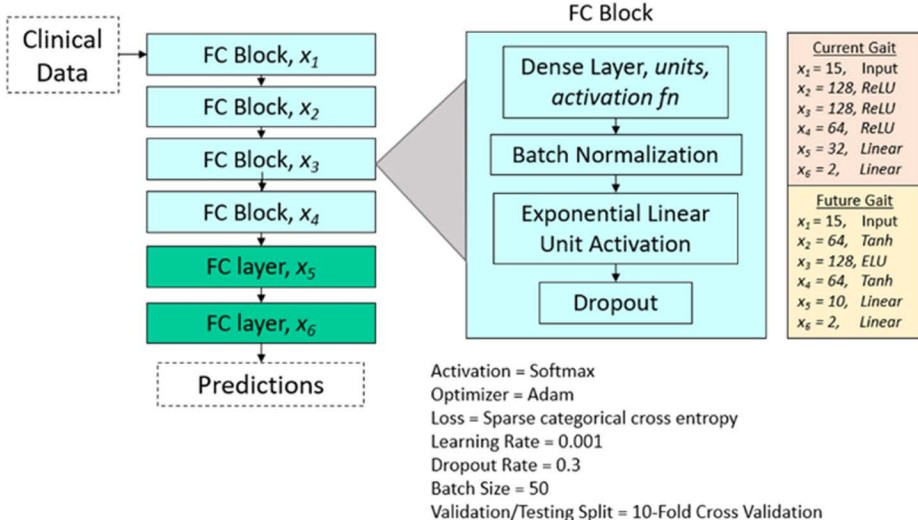

**Fig 1. Schematic of the NN architecture.** Models for current gait prediction and future gait prediction differ in layer size and activation functions. The final two layers, colored green, do not use the batch normalization and the dropout procedure implemented in previous blocks.

These class balancing procedures were applied to the training set within each cross-validation fold. This ensured that the testing set was composed exclusively of non-synthetic samples and that there was no leakage from the training set. Augmentation of each fold also reduced bias, reducing the effect of any outlier synthetic samples. Undersampling of each fold separately ensured that majority class samples containing potentially important or unique information were not removed entirely from the experiment. Data augmentation was not performed on the 10-year classifier for the 0.8 m/s cut-point given the approximately equal number of slow and normative samples in that dataset.

**2.3.1 Random undersampling (RUS).** A manual RUS procedure was used to randomly remove samples from the majority class of the training set until each classifier had an approximately even number of slow and normative samples. Applying RUS individually to each fold rather than to the full training set ensured that normative samples containing potentially important samples were not removed entirely from the experiment.

**2.3.2 Synthetic minority oversampling technique (SMOTE).** SMOTE was introduced as a method to increase the sensitivity of predictor's performance on imbalanced datasets without sacrificing large amounts of data through random undersampling [67]. To implement this, a sample from the minority class is drawn and its $k = 3$ nearest neighbors, determined by Euclidian distance, are identified. A vector is drawn to one of those neighbors from the selected sample and then multiplied by a number between 0 and 1. The resultant vector is then added to the selected sample to create a new synthetic data point [66]. This procedure is applicable only to numeric data. SMOTE-NC is a variation of this which can be applied to nominal data as well. We implemented SMOTE-NC using the Imbalanced Learning Library in Python to achieve an equal class distribution between slow and normative walkers [68].

**2.3.3 Synthetic minority oversampling technique with edited nearest neighbors (SMOTE-ENN).** SMOTE-ENN is a popular class-balancing technique that was developed to combine SMOTE's ability to generate new minority data with the ENN undersampling algorithm [69]. Following generation of new synthetic data points through SMOTE, ENN is applied. Using the same k-nearest neighbors technique, a sample and its neighborhood is identified. If a sample is found to have a different label than the majority of labels in its neighborhood, then all observations are deleted. Repeating this procedure results in equalized classes as well as less overlap between classes.

## 2.4 Sobol sensitivity analysis

We implemented a Sobol index sensitivity analysis to identify the relative importance of clinical variables for determination of current slow gait as well as development of slow gait over a defined timeframe. Sobol sensitivity analysis is a form of variance-based global sensitivity analysis that assesses the degree to which a model's output variance can be attributed to each input variable. The importance of individual variables is assessed by first-order indices, while higher order indices indicate the importance of interactions [70]. We implemented Sobol index analysis using the Sensitivity Analysis Library (SALib) in Python [71,72]. Parameters with resulting Sobol indices >0.05 are considered significant.

## 3 Results

### 3.1 Model performance

**3.1.1 Optimized models.** The best performing models of both the NN and the LR were class-balanced with the RUS method. The final dataset sizes after RUS class balancing are listed in Table 3.

The Youden indices of RUS class-balanced models for the NN and LR analyses are shown in Fig 2. These results, along with sensitivity, specificity, precision, and AUPRC values are listed in Table 4. The best performing classifier was for the 10-year prediction using a 0.8 m/s cut-point, where the NN achieved a sensitivity and specificity of 81.2% and 87.9%, respectively. This performance is similar to that of the LR which achieved sensitivity and specificity in this case of 84.5% and 86.3%, respectively.

**Table 3. Final dataset sizes for each classifier after RUS class balancing.**

| Cut-point | Dataset | Normative | Slow | Total |
|---|---|---|---|---|
| 0.8 m/s | Current | 276 | 276 | 552 |
| | Future: 6 Years | 181 | 181 | 362 |
| | Future: 10 Years | 247 | 245 | 492 |
| 1.0 m/s | Current | 910 | 910 | 1820 |
| | Future: 6 Years | 274 | 274 | 548 |
| | Future: 10 Years | 186 | 186 | 372 |

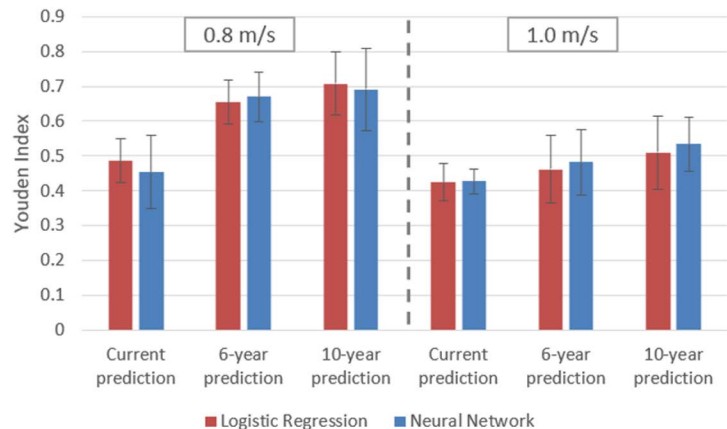

**Fig 2. Youden index for NN (blue) and LR (red) models after class balancing using RUS.** RUS class balancing was not implemented for the 10-year prediction due to the balance in the native dataset. Error bars were determined by the standard deviation across all cross-validation folds.

**3.1.2 Comparison of models with unbalanced classes.** To directly compare the performance of the NN and LR in the unbalanced case, the Youden indices for the NN and LR models without class balancing are presented in Fig 3; these results, along with sensitivity, specificity, precision, and AUPRC values are listed in the Supporting Information, S2 Table. The two distinct model types performed comparably across all cut-points and timeframes, with the NN slightly outperforming the LR in all but one case.

**3.1.3 Exploration of class balancing.** In addition to RUS, we explored two other class balancing techniques, along with the unbalanced case. Results for each classifier before and after data modification with the class-balancing techniques described above are shown in Fig 4a; these results, along with sensitivity, specificity, precision, and AUPRC values are listed in the Supporting Information, S3 Table. For the 10-year prediction with 0.8 m/s cut-point, classes were approximately equal without balancing, which was therefore not performed.

Performance of the LR for each dataset after the various class-balancing techniques are shown in Fig 4b. The Youden index, sensitivity, specificity, precision, and AUPRC values are listed in the Supporting Information, S4 Table. Both the NN and LR models were substantially improved through use of class balancing in all but one case (the 10-year prediction with 1.0 m/s cut-point).

## 3.2 Sensitivity analysis of optimized models

The results of the Sobol index sensitivity analysis for the optimized, RUS-balanced NN and LR models for each dataset are shown in Fig 5. All variables found to be significant (with an index greater than 0.05) are also listed in order of their

**Table 4. Performance metrics for the NN and LR models using RUS class balancing.**

| | Cut-Point 0.8 m/s | | | | | |
| --- | --- | --- | --- | --- | --- | --- |
| | Current Prediction | | 6-Year Prediction | | 10-Year Prediction | |
| | LR | NN | LR | NN | LR | NN |
| Youden index | 0.49 | 0.45 | 0.65 | 0.67 | 0.71 | 0.69 |
| Sensitivity | 0.78 | 0.73 | 0.84 | 0.85 | 0.85 | 0.81 |
| Specificity | 0.71 | 0.73 | 0.81 | 0.82 | 0.86 | 0.88 |
| Precision | 0.17 | 0.17 | 0.50 | 0.51 | 0.86 | 0.88 |
| AUPRC | 0.22 | 0.19 | 0.52 | 0.51 | 0.43 | 0.41 |
| | Cut-Point 1.0 m/s | | | | | |
| | Current Prediction | | 6-Year Prediction | | 10-Year Prediction | |
| | LR | NN | LR | NN | LR | NN |
| Youden index | 0.43 | 0.43 | 0.46 | 0.48 | 0.51 | 0.53 |
| Sensitivity | 0.75 | 0.70 | 0.43 | 0.68 | 0.75 | 0.72 |
| Specificity | 0.68 | 0.73 | 0.72 | 0.80 | 0.76 | 0.81 |
| Precision | 0.42 | 0.45 | 0.53 | 0.59 | 0.87 | 0.89 |
| AUPRC | 0.32 | 0.31 | 0.36 | 0.35 | 0.25 | 0.25 |

RUS class balancing was not implemented for the 10-year prediction due to the balance in the native dataset. Results are shown for all six classifiers. The AUPRC value is the difference between the AUC and the no-skill AUC.

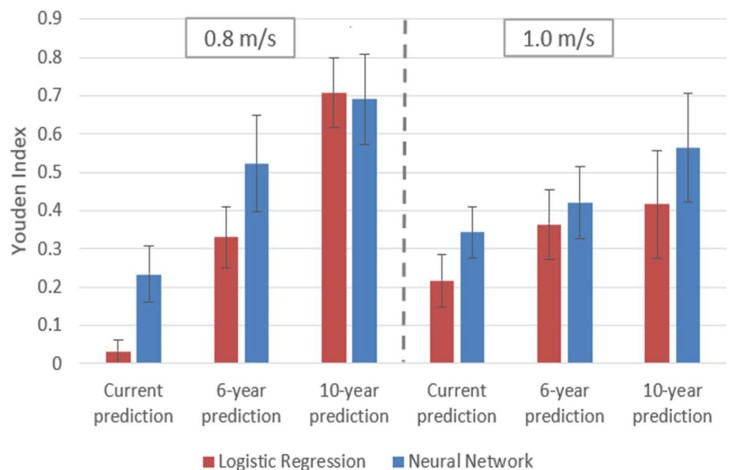

**Fig 3. Youden index values for NN (red) and LR (blue) models without class balancing.** As seen, the NN exhibits overall superior performance. Error bars were determined by the standard deviation across all cross-validation folds.

Sobol index in the Supporting Information, S5 Fig. Across all the classifiers tested except one, age is the strongest predictor, though it is never the only significant predictor.

## 4 Discussion

The purpose of this work was to predict aging-related slow gait and its determinants across various timeframes. The key contributions of this work are: (1) the development of a NN which can successfully predict slow gait with a flexible

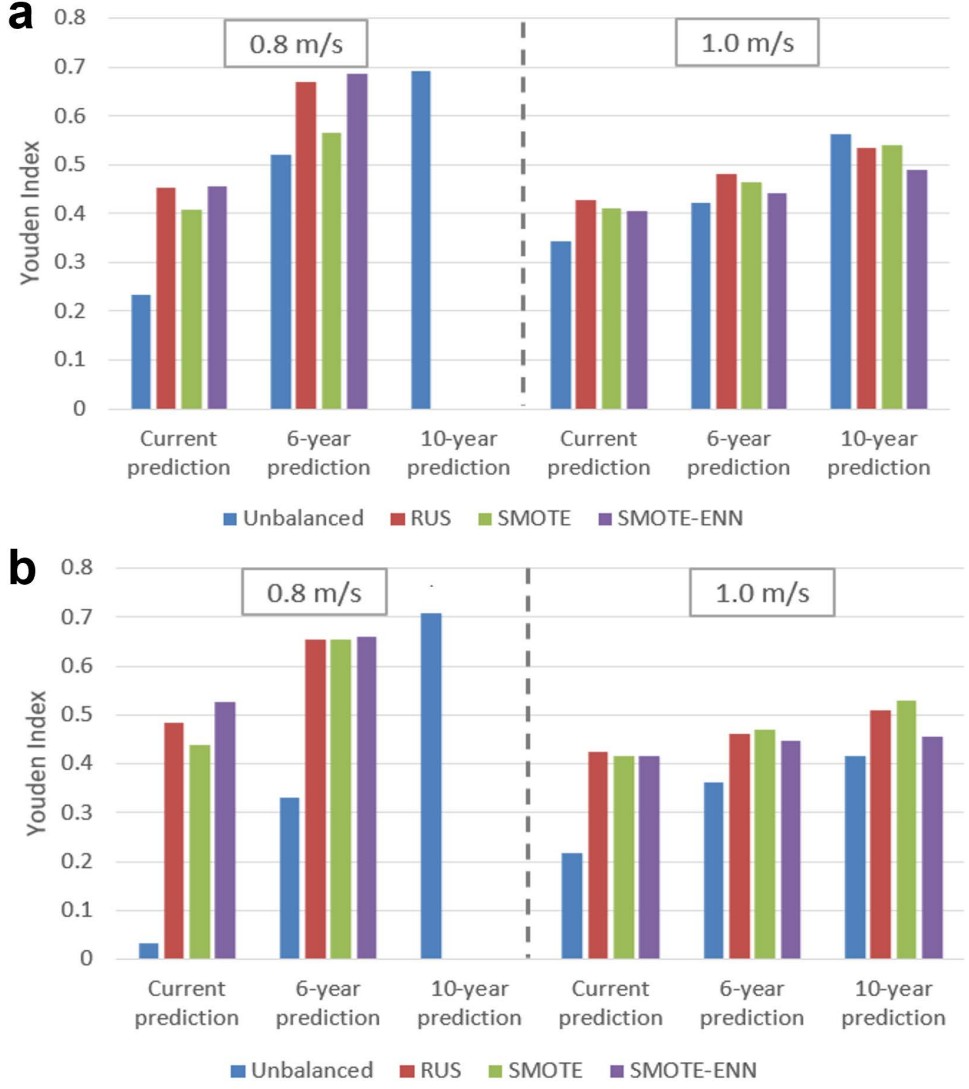

**Fig 4. Youden index values for all class-balancing techniques beside the unbalanced case (blue bars) for the** (a) NN and (b) LR model. Class-balancing techniques resulted in improved Youden indices. Class balancing performed by RUS, SMOTE, and SMOTE-ENN algorithms are represented with red, green, and purple bars, respectively. Class balancing was not implemented for the 10-year prediction due to the balance in the native dataset.

structure permitting several natural extensions of the present work, (2) the demonstration of the comparable or improved performance of the NN relative to a conventional LR, (3) the determination that the key determinants of future slow gait are age, BMI, sleep, and grip strength, and (4) the validation of the fact that class balancing substantially improves performance over models with unbalanced datasets.

## 4.1 Model performance

**4.1.1 Optimized models.** The results for the optimized, class-balanced models (Fig 4 and Table 4) demonstrate that the NN performed as well as or better than the LR. The best performing classifier was for the 10-year prediction using a 0.8 m/s cut-point, where the NN achieved a sensitivity and specificity of 81.2% and 87.9%, respectively. This performance

## 0.8 m/s 1.0 m/s

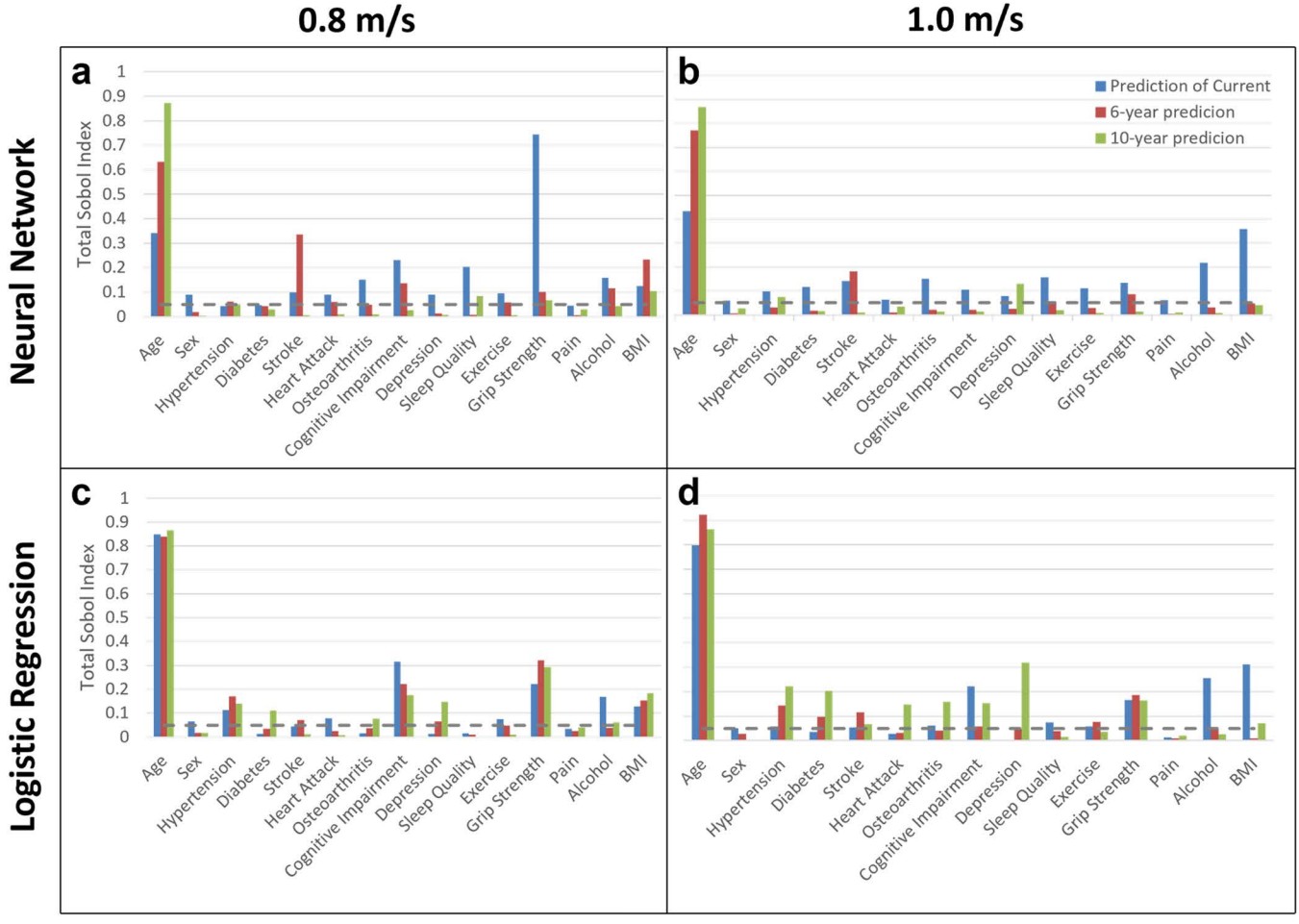

**Fig 5. Sobol indices ranking the relative importance of input variables for model predictions.** Results are shown for the (a) NN with a 0.8 m/s cut-point, (b) NN model with a 1.0 m/s cut-point, (c) LR model with a 0.8 m/s cut-point, and (d) logistic regression model with a 1.0 m/s cut-point. Total Sobol values are shown in blue for prediction of current gait speed, red for prediction of gait speed in 6 years, and green for prediction of gait speed in 10 years. As seen, age is by far the most influential variable in all cases, with the importance of the other variables differing among the four panels.

is similar to that of the LR which achieved a sensitivity and specificity of 84.5% and 86.3%, respectively. This is on a par with other complicated clinical analyses, but clearly encourages further work towards improvement.

Of note, all models performed, at best, with only moderate success; the maximum Youden index, for example, was ~0.7 (for 10-year prediction of slow gait with respect to the 0.8 m/sec cutoff). This motivates the future use of additional input data and perhaps elimination of those that were found not to provide substantial predictive power. Even with comparable performance, NN methods have an important advantage over LR for future work in that they are adaptable to the incorporation of images into further analysis. In contrast, the LR approach can incorporate image-derived metrics, such as regional volumes, but does not allow for use of images themselves. Similarly, NNs can be expanded to incorporate longitudinal data in a formal way, unlike LR.

We found that all the models performed better for future than for current gait prediction, despite the latter's greater training set size (Table 3). This may reflect the particular set of input variables we chose or indicate that the chosen markers reflect physiologic status that emerges clinically only after a period of time. This relationship may even be time

dependent as we also note that the 10-year classifiers slightly outperformed the 6-year classifiers, again independent of the training set size. Finally, we note that classifiers for the 0.8 m/s cut-point outperformed those using the 1.0 m/s cut-point for the equivalent timeframes, suggesting that the more extreme cut-point was more easily identified as abnormal by this set of input variables.

While the performance difference between the NN and the LR was minimal in the present study, the additional flexibility afforded by the NN and its ability to capture nonlinear relationships may prove to be crucial for future studies of gait and related physical outcomes. In particular, the use of NN allows for the addition of more complex inputs such as images and other non-tabular data.

**4.1.2 Comparison of models with unbalanced classes.** Fig 3 shows that when using unbalanced data, across all cut-points and timeframes, the NN slightly outperforms standard LR except in the case of the 10-year prediction using a 0.8 m/s cut-point. However, as in the balanced data case, the model types perform comparably. The relatively weak performance of the current and 6-year predictions for both cut-points is largely attributable to low sensitivities seen in S2 Table in the Supporting Information. Such sensitivity values are not surprising given the extreme class imbalance observed for the corresponding data (Table 1). The NN exhibited substantially greater sensitivity than LR in these cases, suggesting the comparative resiliency of the NN approach to data imbalance. Nevertheless, the sensitivity and Youden index are still poor overall, motivating the exploration of class-balancing techniques to improve sensitivity.

Issues such as class balancing have come to the fore with the recent explosion of interest in DL and are known to virtually all practitioners. However, such methods have been much less recognized in the context of more traditional analytic methods such as LR. In that sense, the conventional NN approach incorporating class balancing (Fig 2), greatly outperforms the conventional LR approach in which no class balancing is performed (Fig 3). These results highlight therefore the great power of multivariate linear analysis in conjunction with more modern considerations related to dataset structure. This leaves open the question of why the theoretically more limited LR approach exhibited performance on par with the NN after class balancing. Evidently, for the variables selected and over the range of values studied, linear effects capture the dominant biomarkers of gait without the need for nonlinear modeling.

**4.1.3 Exploration of class balancing.** As indicated by the Youden index results shown in Fig 4, both the NN and LR models were substantially improved through use of class balancing in all but one case. Increases in Youden index were most evident in models using a 0.8 m/s cut-point, for which class imbalance was more severe, but those with a 1.0 m/s cut-point also saw improvements. Increases in Youden index were driven by large increases in sensitivity as seen in the Supporting Information, S3 and S4 Tables. These increases were most apparent for current timeframe predictions, particularly with LR and for the 0.8 m/s cut-point for which the model was almost entirely unable to identify slow walkers. The current timeframe datasets had the greatest imbalance in the native data (see "% Slow" column in Table 1), and it was in fact for these that class balancing led to the largest increases in performance. These results strongly support the use of class balancing in related studies, particularly since our classifiers were trained on balanced data and tested on natural ratios; this is a more difficult classification task than balancing both training and testing sets.

The one model that did not benefit from class balancing was the NN designed to predict gait speed class in 10 years with the 1.0 m/s cut-point. We attribute this to the fact that this dataset was already well-balanced, actually containing more slow than normative walkers, in contrast to all other datasets used.

The three class balancing methods, RUS, SMOTE, and SMOTE-ENN, exhibited overall similar performance. We therefore selected RUS for class balancing, given its simplicity and the fact that it does not require synthetic data. It is interesting that the simplest approach, RUS, performed as well as the more sophisticated techniques of SMOTE and SMOTE-ENN. In this context, we note characteristics of SMOTE that may have limited its effectiveness in our study. Overlap between classes is particularly problematic for SMOTE, given that it blindly generalizes the region of a minority class without consideration of nearby samples of the majority class [66]. This strategy is particularly problematic in the case of highly skewed class distributions since the minority class tends to be sparse with respect to a majority class, thus

resulting in a greater chance of class mixture [73]. SMOTE can also exhibit limited performance in the setting of multiple feature inputs; it becomes much more difficult to generate a representative sample of new data in higher dimensions. High dimensionality also gives rise to the phenomenon of hubness, in which a small number of points are overrepresented in the selection of nearest neighbors [74,75]. Given the efficacy of class balancing in this study, further investigation into the utility of other available methods may be warranted.

## 4.2 Sensitivity analysis

Sensitivity analysis was captured by the Sobol indices plotted in Fig 5 and listed in S5 Fig in the Supporting Information. In the best performing NN classifier (10-year prediction with 0.8 m/s cut-point; Fig 5b, green bars), we found five significant predictors with the strongest being age, BMI, and sleep quality. For the LR model, nine significant predictors were identified, with the strongest being age, grip strength, and BMI. Four of the five variables that were identified as most predictive in the NN analysis were also most predictive in the corresponding LR model. The remaining variable, sleep quality, may enter in a nonlinear fashion that required the NN to uncover. This motivates further comparative studies to characterize the nature of the relationships between clinical variables and slow gait outcomes.

Other methods for assessing sensitivity to particular variables were considered. These included logistic regression coefficients, Shapley Additive Explanations (SHAP), and Local Interpretable Model-agnostic Explanations (LIME). Sobol Index analysis was chosen due to (1) the need to have a single consistent metric across both model types, (2) its global ability to measure sensitivity across the entire input space, and (3) its applicability to nonlinear relationships.

### 4.2.1 Trends across time, cut-point, and model type.
We found that across all the classifiers tested except one, age is the strongest predictor. For the current prediction with 1.0 m/s cut-point, grip strength was a stronger predictor than age. Of note, this was also the lowest-performing classifier. The dominance of age as a predictor was as expected, but we found that its influence was in all cases modulated by other significant predictors.

For the NN, age was less dominant as a predictor for current than for future gait. Consistent with this, there were a greater number of significant variables in these analyses. In fact, for current prediction with a 1.0 m/s cut-point, every input variable was found to be significant. In contrast, for the 10-year prediction with the same cut-point, only three variables were found to be significant: age, depression, and hypertension. This result suggests that a wider array of the chosen variables exhibit a substantial predictive power for current slow gait as compared with future slow gait, which may reflect certain effects becoming dominant over time. In particular, osteoarthritis, diabetes, pain, and sex are significant for current but not for future predictions. One hypothesis for the differences across time is that certain metrics exhibit a more delayed influence on the aging trajectory of a subject. For example, smoking or consistent poor sleep quality presumably correlates to higher morbidity over longer time scales, while poor strength influences gait contemporaneously.

For the future prediction NN classifiers, more variables were found to be significant for the 0.8 m/s cut-point than the 1.0 m/s cut-point. Grip strength appeared as a significant variable in all but one classifier (10-year prediction with 1.0 m/s cut-point). Sleep quality, BMI, stroke, and hypertension appeared in four classifiers as significant predictors. Interestingly, for the 6-year prediction, age and stroke were the strongest determinants across both cut-points, but stroke did not appear among the predictive variables for the 10-year timeframe.

In this study, we performed direct comparisons of two different model structures, NN and LR. This allowed us to capture model dynamics and consider the key determinants of healthy aging within this dataset without model constraints. We found a similar set of dominant predictors for the LR (Figs 5c, 5d, and S6) as for the NN (Figs 5a, 5b, and S5). In particular, age, grip strength, BMI, and hypertension appeared as significant most frequently across both model types. Further, all four of these were found to be significant in our best performing models (10-year prediction with 0.8 m/s cut-point for both NN and LR). On the other hand, stroke and sleep quality were found to be significant more consistently in the NN than in the LR. This may indicate a stronger non-linearity in the relationship between these variables and gait speed.

Additionally, we found that more variables appeared as significant for the LR model as compared to the NN for the 10-year prediction, but fewer were significant for the LR for current prediction. We also found that the significant predictors in the LR were more consistent across classifiers than in the NN. Age, grip strength, MMSE score, and hypertension appeared as significant in every LR classifier, with age and grip strength among the top three factors in four classifiers and among the top five factors across all LR classifiers. BMI was found to be significant in five of the six LR classifiers, while stroke was found to be significant in four. Interestingly, diabetes and depression only appeared as significant in future prediction classifiers, not those of current timeframe; this may indicate the importance of these variables over time. Pain was the only variable that never appeared as significant in any of the LR classifiers.

While our methodology does not indicate causality versus correlation, the fact that future gait speed can be predicted by models which include modifiable risk factors may prove to be of substantial clinical utility. For example, BMI and grip strength were consistently found to be significant according to Sobol index analysis and can clearly form the basis for therapeutic intervention and guidance.

**4.2.2 Consideration of survey data and strongest predictors.** We conducted several additional sensitivity studies with the NN to further define the impact of certain variables. These investigations were confined to the 10-year prediction of the 0.8 m/s classification cut point using balanced data; see S6 Fig.

Our variables included both self-reported survey data and objective, quantifiable data. The limitations of self-reported data have been well-recognized [76]. We therefore investigated the specific contribution of these variables to our models by performing a NN analysis using only sex and the four non-categorical (non-survey) variables of age, BMI, grip strength, and cognitive score as input variable. This resulted in comparable performance with a Youden index of 0.68 as compared to the Youden index of 0.69 for the original dataset of all 15 variables. These results are shown in the Supporting Information, S5 Table. This indicates that the categorical variables do not provide substantial additional predictive power to the model.

Since age was the dominant predictor as indicated by its large Sobol index in all models, we also developed a NN model with age and sex as the only input variables. The resulting Youden index of 0.65 indicated a somewhat degraded performance as compared to the all-variables result of 0.69. The decrease was due to lower sensitivity, 0.764, as compared to the all-variables sensitivity of 0.812. To further investigate the role of dominant variables, we trained a NN model using all variables except age. As expected based on Sobol index analysis, we found a substantial decrease in the Youden index, from 0.69 to 0.49. While performance indeed worsened through omission of age, it is noteworthy that the other variables alone still achieved decent performance for gait classification.

Previous investigators have established a strong correlation between height and gait speed, unsurprisingly, especially when examining current timeframes [77]. However, when we swapped height for BMI in our models, we encountered no discernable difference in the model performance. Thus, we did not include height in our classifiers.

We conclude that the 15 original variables formed a reasonable input set. While performance was essentially maintained when categorical variables were removed, these results support their use in that the self-reported survey data did not negatively impact performance. The results obtained when omitting age provided evidence that multiple input variables are indeed important for gait speed prediction.

## 4.3 Gait speed prediction in the literature

Previous analyses of gait speed as it relates to aging trajectories differ from the current study and present several limitations. A number of studies evaluate gait speed as a model input, rather than as an outcome [4,28–33]. In fact, that line of investigation demonstrates the importance of gait speed as a measure of health status and provides the motivation for our study. Most studies that, like ours, evaluate the determinants of gait speed examine only predictors of current status, rather than predicting future aging trajectories [21–27]. Our inclusion of future predictions as a fundamental target of our work was motivated by the potential clinical utility of aging trajectory prediction and predicated upon the idea that health

factors may show increasing impact over time. Indeed, we found notably better predictive ability for both the NN and the LR models for future gait than for current gait.

**4.3.1 Data considerations.** Our chosen variables are based on those identified in other studies, and most closely following the important work of Verghese et. al [34]. Other investigators have included similar sets of modifiable risk factors and medical conditions, [34–36] with some also incorporating or focusing on cognition [37,38]. In addition, inflammatory markers, [39] body composition, [2] and brain volumes [35] have all been incorporated into gait studies. While lower extremity strength would seem an influential factor in the analysis of gait determinants, few studies have incorporated it [2,23,25]. This may be due to its presumed high correlation with the gait outcome itself, or with the relative difficulty and lower reproducibility in measuring it as compared to hand grip strength. Of note, previous work has shown that grip strength contributed more than lower-extremity strength to variance in walking speed [78].

In spite of this literature, there remains a knowledge gap regarding the relationship between gait speed decline and a range of potentially determinative factors. Although in the present case LR performed on par with the more flexible NN model, we conjecture that these more flexible DL architectures will prove to be better able to capture the influence of complex datasets, including evaluation of longitudinal variables and raw imaging data. An important part of our effort was to develop a DL model and to establish its performance on well-studied variables, so that with further developments, we will be able to more deeply exploit the full potential of the extensive, longitudinal BLSA database.

A further advance in the present work is the evaluation of two clinically relevant gait speeds, defined as the point of severe mobility disability (0.8 m/s) and the speed below which the risk of mortality doubles (1.0 m/s). In contrast, a number of previous studies employ very low cut-points which capture only the most severely impaired subjects [27,34].

**4.3.2 Model considerations.** Previous work on future gait speed prediction has implemented only linear statistical models, including mixed-effects models, [2,37,38] hierarchical regressions, [35]. Poisson regressions, [34] and cross-sectional analysis [39]. Similarly, current gait speed prediction models have also mainly employed linear regressions [23–26,58] and cross-sectional analyses, [22] along with logistic regressions, [21,27] although support vector regression [79] and decision tree analysis [48] have also been explored. We sought to introduce the NN approach to this important problem, given the complexity of human biochemistry and physiology. With numerous contributing and overlapping factors to consider, AI techniques, such as NNs, offer a potentially superior tool for the study of gait speed decline and aging trajectories. Deep learning models allow for the simultaneous consideration of multiple patient factors, and thus tend to be more suitable than classical statistical methods for problems involving large numbers of predictors by accounting for their combined influence and accommodating the inclusion of complex data such as images, clinical data, and other biomarkers and health [80,81]. Thus, our work represents a promising new application of NNs and, in addition to our biomedical results, serves as a pilot study towards exploration of more complex input variables including images and longitudinal data.

**4.3.3 Sensitivity analysis.** Previous studies have used a variety of performance metrics, making direct comparisons difficult. Several of these, like ours, have found BMI, [34,35] grip strength, [34,35] and cognitive impairment [34,37] to be significant predictors of gait. However, other studies found these variables not to be significant [2,38]. Verghese et. al also found exercise and pain to be significant [34]. While Pinter et. al identified age as a strong determinant [35], we note that most modeling studies use age as a covariate or for stratification, instead of investigating its influence directly as we did. Though there is not a strong, clear consensus, there is overall support in the literature for our findings that age, grip strength, and BMI are among the strongest determinants of future gait speed. Sleep quality also emerged as an important predictor in some of our models, as did cognitive impairment and exercise. As compared to previous studies, we evaluated model performance over a longer timeframe (six and ten years, rather than three-to-four years), allowing us to capture longer-term effects. This is especially important for the BLSA dataset, which is derived from a relatively healthy cohort in which slow gait may develop over longer time scales.

Previous investigators have identified additional significant predictors of future gait speed decline include vision loss, [34] balance metrics, [34,36] brain volume and white matter hyperintensity change, [35] baseline gait speed, [36] difficulty with activities of daily living, [36] frailty, [36] reaction time, [36] thigh intermuscular fat area, [2] and inflammatory markers [39] Beavers et. al even found the longitudinal change in thigh fat and thigh muscle to be a strong predictor [2]. These findings further motivate the NN approach with application to a richer set of input variables.

## 4.4 Limitations

The main limitations of our study center around available data and potential biases. The BLSA cohort consists entirely of volunteers, who are overall healthier and more educated than the general population. Further, the cohort is largely limited to a catchment area of approximately 3 hours driving distance from Baltimore, MD. Other NIA-sponsored study populations, notably the HANDLS cohort which represents a population with greater diversity, [82] would therefore represent important opportunities for further study. Study investigators have targeted specific populations in their recruitment in an effort to achieve more balance in the dataset. Another limitation is that gait speed was not measured in the BLSA until 2004. This may introduce some bias in the upper age groups as, for example, the older individuals in the cohort represent exceptional agers whose initial gait speed measurement would have taken place at an advanced age. To explore the influence of these potential biases on our models, we investigated trends in several variables across the BLSA database and found the resulting dataset drift to be minimal (S2 and S3 Figs in the Supporting Information).

The BLSA dataset includes participants with a wide range of non-debilitating medical conditions. This increases the generalizability of our models and emphasizes that gait speed represents an outcome metric which integrates the function of multiple organ systems. As with predictions in any heterogeneous sample, this approach may limit model performance in terms of numerical metrics, while nevertheless appropriately addressing the overall question of the determinants of gait speed in the study population. Future work may therefore include subgroup analysis with participants stratified by specific underlying pathophysiology, resulting in a very different, but more targeted study. For example, prediction of gait speed trajectories for subjects with neurodegenerative disease may be of substantial clinical value. In any event, given the dataset requirements of NN training, more targeted studies with narrowly specified entrance criteria may require a very different recruitment strategy, given the largely normative aging population represented by the BLSA.

There were also certain limitations to our study based on available data. We were unable to include a variable indicating number of recent falls since fewer than half of recorded timepoints incorporated this measurement. In addition, ~ 15% of the datapoints for MMSE scores required imputation. The use of imputation, and the choice of the median for replacement, while a common method, may introduce a small degree of bias, and evaluation of alternative methods for imputation of missing data may be of interest in subsequent work. Here, we implemented a conventional approach and did not further investigate these issues. Another potential limitation is that gait speed was measured by trained observers rather than by truly objective operator-independent methods. However, this approach has been standardized and used extensively in the BLSA and other studies, with high test-retest reliability (ICC > 0.87). We also note that the majority of our input variables were available only as survey data, which may be less reliable than data extracted from medical records. However, BLSA survey data may be more reliable than most, given the relatively healthy status of BLSA participants and the attribution of unreliable survey data to impaired cognition and memory. Future work should consider more objective measures of the survey data included here.

We chose to include grip strength from only the right-hand side. For most participants, both right and left grip strength was included in the dataset and many also had their dominant hand noted. While we would not anticipate a significant impact on the model performance given the strong correlation we found between the right and left grip strength (Pearson coefficient = 0.74), future work could explore any impact of using the mean of the two grips or the dominant hand. Note also that using a single hand makes the model more generalizable to other datasets and more accessible for clinical implementation.

Extensions of the current study also include additional variables of interest that were excluded here due to the dataset limitations described in Section 2.1.1. These include inflammatory markers, body composition, brain volume, frailty, falls (balance), and vision. A great deal of current clinical research is also focused on social determinants of health, which are emerging as key variables in well-being. These will be key predictors to explore in future work as more of these determinants (e.g., race, income, education) are prioritized and recorded on a consistent basis.

The availability and consistency of the dataset will be a key limiting factor in the proposed future studies of longitudinal data. Initial attempts to implement a recurrent neural network (RNN) study of the present data in order to establish longitudinal effects have met with limited success, likely due to the strict requirements of an RNN for time point repeatability for a sufficiently large sample size. Additional approaches will be explored, given the importance of understanding the determinants of gait speed and its trajectory in aging.

## 5 Conclusion

This study identified determinants of slow gait in the aging population across several timeframes, noting that age, BMI, sleep, and grip strength are key determinants. In addition, we established a NN model for the exploration of aging-related gait speed decline which performs comparably to or better than a conventional logistic regression model. This structure includes the potential for incorporating additional health measures, more complex inputs including images, and longitudinal data. Our future work will incorporate additional variables with potentially non-linear relationships to gait speed including brain myelination patterns, muscle bioenergetic indices obtained with $^{31}$P magnetic resonance spectroscopy, blood or urine biomarkers, leg dominance, and gait biomechanics metrics [22]. A more systematic and extensive exploration of factors such as the social determinants of health and patient-reported outcome measures (PROMs) may provide additional understanding of gait determinants. Some of these advances will be modelled on our previous work developing a CNN+RNN deep learning model for Alzheimer's disease classification [83,84].

The novel use of a NN for this purpose, and demonstration that it performs as well or better than a regression, justifies further expansion of this model to the consideration of more health measures, more complex inputs, and the incorporation of longitudinal information. While the NN and LR demonstrated similar performance in the present study, justifying the use of either approach, our future work will focus on the NN model due to its ability to capture nonlinear relationships, incorporate images, handle multiple types of variables at once, and more easily integrate longitudinal information. Development of this model and evaluation of its dominant predictors advances the understanding of aging trajectories as they relate to gait speed. Further development of these models may assist in complex decision-making in the clinical setting, aiding in the development and implementation of aging interventions which can improve patient quality-of-life and population health-span.

## Supporting information

**S1 Fig. Pearson correlation coefficient for each pair of input variables.** The scale bar on the right shows the color coding from red (strongly positively correlated) to blue (strongly negatively correlated).
(TIF)

**S1 Table. Key demographics of our dataset comparing the included and dropped data from the BLSA database.** Values are median +/- standard deviation. Bold text indicates values that were found to be significantly different.
(TIF)

**S2 Fig. Analysis of potential dataset drift in the year of visit for all BLSA subjects used in this study.** The lack of a clear trend across the means suggests minimal dataset drift over the year of visit.
(TIF)

**S3 Fig.**   a) Analysis of potential dataset drift in the year of birth for all BLSA subjects used in this study. The dotted line indicates a linear regression fit with regression coefficient of 0.0082. b) Comparison to a regression fit of age versus gait speed (decreasing left to right) with coefficient of 0.0083. The similar coefficients and shape of the data indicate minimal dataset drift due to year of birth.
(TIF)

**S4 Fig.  An example learning curve for the NN training on 10 Year 0.8 m/s classifier.** The blue line indicates the loss of the training set, and the orange line represents the loss of the validation set. These loss curves show minimal overfitting after hyperparameter tuning.
(TIF)

**S2 Table.  Performance metrics for the classifiers trained with unbalanced datasets.** The AUPRC value is the difference between the AUC and the no-skill AUC.
(TIF)

**S3 Table.  Performance metrics of classifiers using the NN with various balancing techniques.** The AUPRC value is the difference between the AUC and the no-skill AUC. RUS = Random Undersampling, SMOTE = Synthetic Minority Oversampling Technique, ENN = Edited Nearest Neighbors. The asterisk indicates the classifier that did not need class balancing techniques because the data was already balanced.
(TIF)

**S4 Table.  Performance metrics of classifiers using the LR with various balancing techniques.** The AUPRC value is the difference between the AUC and the no-skill AUC. RUS = Random Undersampling, SMOTE = Synthetic Minority Oversampling Technique, ENN = Edited Nearest Neighbors. The asterisk indicates the classifier that did not need class balancing techniques because the data was already balanced.
(TIF)

**S5 Fig.  Total order Sobol indices above 0.05 (significance) for each classifier using the NN.** The color scale is shown in the inset, where darker colored cells indicate variables that are found to be significant more frequently across classifiers.
(TIF)

**S6 Fig.  Total order Sobol indices above 0.05 (significance) for each classifier using the NN.** The color scale is shown in the inset in S5 Fig, where darker colored cells indicate variables that are found to be significant more frequently across classifiers.
(TIF)

**S5 Table.  Results of the sensitivity analysis performed to study the influence of various input variables.** A: 15 Original inputs, B: Original inputs with height in place of BMI, C: Age and sex alone, D: Age, sex, BMI, grip strength, cognitive score (all the quantitative original inputs) E: Original inputs except age.
(TIF)

## Acknowledgments

We would like to thank all those who contributed to and guided this research =, including the Jeraj (Image Guided Therapy) Lab Group at University of Wisconsin – Madison who offered input into the deep learning aspects of the work. We are additionally grateful to all those who have helped accumulate the vast amounts of data in the BLSA database and to Elango Palchamy at the NIA who helped us access and curate the data for our use.

## Author contributions

**Conceptualization:** Alison Deatsch, Jonathan Palumbo, Qu Tian, Eleanor Simonsick, Luigi Ferrucci, Robert Jeraj, Richard G. Spencer.

**Data curation:** Alison Deatsch, Michael McKenna, Jonathan Palumbo, Eleanor Simonsick, Luigi Ferrucci.

**Formal analysis:** Michael McKenna, Jonathan Palumbo.

**Investigation:** Richard G. Spencer.

**Methodology:** Alison Deatsch, Michael McKenna, Jonathan Palumbo, Richard G. Spencer.

**Project administration:** Richard G. Spencer.

**Resources:** Luigi Ferrucci, Robert Jeraj, Richard G. Spencer.

**Supervision:** Alison Deatsch, Richard G. Spencer.

**Validation:** Eleanor Simonsick.

**Writing – original draft:** Alison Deatsch, Michael McKenna, Richard G. Spencer.

**Writing – review & editing:** Alison Deatsch, Qu Tian, Richard G. Spencer.

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
