## [Decision Letter · Decision Letter 0]

Dear Dr. Deatsch,

Thank you for submitting your manuscript to PLOS ONE. After careful consideration, we feel that it has merit but does not fully meet PLOS ONE’s publication criteria as it currently stands. Therefore, we invite you to submit a revised version of the manuscript that addresses the points raised during the review process.

We look forward to receiving your revised manuscript.

Kind regards,

Esedullah Akaras

Academic Editor

PLOS ONE

Journal Requirements:

Reviewers' comments:

Reviewer's Responses to Questions

**Comments to the Author**

1. Is the manuscript technically sound, and do the data support the conclusions?

Reviewer #1: Yes

Reviewer #2: Partly

Reviewer #3: Yes

2. Has the statistical analysis been performed appropriately and rigorously?

Reviewer #1: Yes

Reviewer #2: Yes

Reviewer #3: Yes

3. Have the authors made all data underlying the findings in their manuscript fully available?

Reviewer #1: No

Reviewer #2: No

Reviewer #3: No

4. Is the manuscript presented in an intelligible fashion and written in standard English?

Reviewer #1: Yes

Reviewer #2: Yes

Reviewer #3: Yes

Reviewer #1: I congratulate the authors for their work. Considering that the decrease in walking speed is an indicator of mortality for elderly individuals with chronic diseases, your work is very valuable.

However, I have a few suggestions and questions.

The intended use of deep learning and NN in elderly individuals with chronic diseases should be detailed in the introduction section.

The fact that the walking speed of the individuals was not measured with an objective equipment can be stated as a limitation.

Only the right side was used for grip strength. Domimancy was not taken into concern?

Measurements of participants' walking speed over time may vary because of the individuals performing the measurements. it should be stated how this standardisation was achieved. Was interrater reliability performed? ICC value ?

In your study, age, grip strength and BMI were reported as major predictors similar to the results in the literature. LR and NN results are also similar. In this case, the superiority of NN or the reason for its preference is insufficient. This section needs to be supported by the authors.

Reviewer #2: This study addresses a highly relevant topic in the field, contributing valuable insights that enhance our understanding of gait and its determinants on aging. The authors have tackled an important question with a well-structured approach, and their findings have the potential to inform both clinical practice and future research. The thoroughness of the methodology and the depth of analysis further strengthen the study’s significance, making it a noteworthy addition to the existing literature. Nevertheless, I believe that implementing the suggested revisions will further enhance the quality and impact of the study, strengthening its contribution to the field.

Intro

1. The authors highlight the need for larger datasets in deep learning analyses and provide sample sizes from previous studies (108, 239, 746, 1901) as examples. However, no references are provided to support these figures. Including citations for these studies would enhance transparency and allow readers to verify the source of this information. (Line 81-86)

2. The introduction would benefit from a clearer explanation of how the present study differs from previous research and which specific gap in the literature it aims to address. For instance, the statement 'Several attempts have been made using statistical models to predict gait speed changes from a narrow set of potential predictors' suggests prior work in this area. However, it would be helpful if the authors explicitly outlined how their study expands upon or improves these past efforts. Additionally, specifying the professional groups (health workers, physiotherapist, physician etc.) that could benefit from these findings would provide readers with a clearer understanding of the study’s relevance and impact.

Methods

3. The primary aim of this study appears to focus on the relationship between gait speed, aging, and mortality. However, gait speed can be influenced by a wide range of factors, including orthopedic, neurological, and other medical conditions. Wouldn't it be more informative to stratify the analysis by different subgroups to account for these variations? Discussing the potential impact of such factors and whether a subgroup analysis could enhance the findings would strengthen the study's interpretation and applicability.

4. I acknowledge that biostatistics and methodological details can sometimes be complex to fully follow, so I apologize if I have missed anything. That being said, I would like to raise a few points regarding the chosen cut-points and the study population. The authors have established clinically relevant cut-points for gait speed (0.8 m/s and 1.0 m/s) based on prior literature. However, gait speed thresholds may vary across different populations, particularly in individuals with neurological conditions such as Parkinson’s disease or stroke. Were disease-specific variations in cut-points considered when applying these thresholds to the study population? If not, this could be a limitation, as the same cut-points may not be appropriate for all individuals.

5. Additionally, the study utilized gait speed data from the BLSA cohort, where measurement intervals varied by age. Since older individuals had more frequent assessments, could this have biased the long-term predictive model? Were adjustments made to account for the potential overrepresentation of gait speed decline in older participants? A discussion on how these factors may have influenced the results would strengthen the study’s findings.

6. It would be better for the readers if the authors explained a point about the measurement and use of grip strength in their analysis. In the table describing input variables, grip strength is reported in kilograms (Table 2). However, it is specifically labeled as "hand grip muscles right (kg)." Could you clarify whether only the right hand was measured and analyzed? If so, what was the rationale for not including the left hand or using an alternative approach such as the mean of both hands or the dominant hand? Additionally, considering that grip strength was identified as a significant factor in the Sobol index analysis and highlighted as a basis for therapeutic intervention, do you believe that using only the right hand impacts the generalizability of your findings? If left-hand data were available, would incorporating it alter your results? Clarifying these points would strengthen the methodological transparency and the applicability of your findings.

7. Previous studies (e.g., Sadeghi et al., 2000) suggest that lower limb dominance plays a role in gait biomechanics, with the dominant limb often contributing more to propulsion while the non-dominant limb provides stability. Considering this, do you think assessing lower limb dominance could enhance the interpretation of gait speed determinants in your study?

Discussion

8. The discussion would benefit from a more detailed justification of why grip strength was emphasized as a key predictor of gait speed rather than more directly related lower-limb strength measures. While grip strength has been associated with overall strength and function, gait speed is likely more directly influenced by lower-limb muscle strength. Addressing this distinction and discussing potential reasons for the focus on grip strength over leg strength measures would provide greater clarity for readers.

9. The authors have considered multiple gait-related parameters in their predictive model. However, in clinical and health-related research, patient-reported outcome measures (PROMs) are frequently used to capture the patient's perspective. PROMs can provide valuable insight into how individuals perceive their mobility, fatigue, pain, or fear of falling, which are factors that could influence gait patterns over time. Did the authors consider incorporating PROMs while examining the determinants of slow gait? If not, this could be a potential limitation of the study, as relying solely on objective gait parameters may overlook important subjective experiences that contribute to mobility decline. Including PROMs in future research could enhance the model's generalizability and provide a more comprehensive understanding of aging-related gait changes.

Reviewer #3: This manuscript presents a valuable contribution to the field of aging and mobility research by exploring predictive modeling techniques for slow gait, a key biomarker of health and longevity. By leveraging data from the Baltimore Longitudinal Study of Aging (BLSA), the study compares the performance of a deep learning neural network (NN) with traditional logistic regression (LR) models in predicting current and future slow gait at different timeframes (6-year and 10-year). Additionally, the study identifies key determinants of gait decline, such as age, BMI, sleep quality, and grip strength.

The study is well-motivated and methodologically rigorous, making a compelling case for integrating machine learning into aging research. However, some methodological and analytical aspects require further clarification or justification:

Strengths________________________________________________

One of the key strengths of this study is its innovative application of deep learning to predict aging-related slow gait. While gait speed has been extensively studied as a predictor of health outcomes, the use of a neural network represents a novel approach that could potentially capture complex, nonlinear relationships between predictors and mobility decline. Additionally, by benchmarking the NN against logistic regression, the authors provide a robust comparative analysis, which strengthens the validity of their results.

The study also benefits from its use of a well-established longitudinal dataset (BLSA), which enhances the reliability of the findings. The long-term follow-up (6 and 10 years) is particularly valuable, as it allows for a more comprehensive understanding of mobility decline over time. Few studies have attempted to predict future slow gait over such an extended period, making this study particularly relevant for aging research.

Another commendable aspect of the study is its attention to data imbalance, a common issue in clinical datasets. By applying various class balancing techniques (RUS, SMOTE, SMOTE-ENN), the authors effectively address the skewed distribution of slow versus normative walkers. This methodological rigor enhances the study’s robustness and provides useful insights into best practices for handling imbalanced datasets in clinical prediction models.

Finally, the study has clear clinical relevance, as it focuses on clinically meaningful gait speed cut-points (0.8 m/s and 1.0 m/s). These thresholds align with established research on mobility disability and mortality risk, ensuring that the findings are directly applicable to clinical decision-making. The identification of modifiable risk factors (e.g., BMI, grip strength, sleep quality) further underscores the study’s potential impact, as these variables could inform targeted interventions for preventing mobility decline.

Areas for improvement and suggested refinements (Minor Revisions)_______________________

While this study presents a strong and well-motivated analysis of aging-related slow gait prediction using deep learning and logistic regression, there are some methodological and conceptual aspects that would benefit from clarification and refinement. These do not require major changes to the core analysis but would enhance the transparency, interpretability, and generalizability of the findings.

- One of the key strengths of this study is its comparison between neural networks (NN) and logistic regression (LR). However, the results indicate that NN performs only marginally better than LR, raising the question of whether the added model complexity is necessary. While the authors state that NN allows for capturing nonlinear relationships, there is no strong evidence that such relationships exist in this dataset.

Suggested Improvement: A brief discussion on whether nonlinear interactions between predictors were observed (or theoretically expected) would strengthen the justification for using NN. If applicable, referencing studies that have successfully demonstrated nonlinear patterns in similar aging-related predictions would provide useful context.

- Deep learning models are susceptible to overfitting, especially when applied to datasets with a relatively small sample size (1,363 participants). The manuscript mentions the use of dropout layers, but there is no explicit discussion of other overfitting mitigation strategies, such as hyperparameter tuning, cross-validation, or external validation.

Suggested Improvement: A short statement on whether techniques such as k-fold cross-validation or regularization methods were used would provide confidence in the model’s generalizability. If external validation was considered but not performed, a mention of this as a future step would clarify the scope of the current analysis.

- A common challenge with deep learning models is their black-box nature, which limits clinical interpretability. While the study employs Sobol sensitivity analysis to rank predictor importance, this approach does not fully address the need for clinically meaningful explanations of how individual variables contribute to predictions.

Suggested Improvement: A brief comparison between the Sobol index results and logistic regression coefficients would help readers understand whether the NN model identifies similar key predictors as traditional methods. Additionally, a sentence or two acknowledging the potential use of SHAP (Shapley Additive Explanations) or LIME (Local Interpretable Model-agnostic Explanations) in future research would be beneficial.

- The study replaces missing values with the median, but the rationale for this choice is not discussed. This is particularly relevant given that 15% of MMSE scores (cognitive impairment) were missing, which could introduce bias.

Suggested Improvement: A brief justification for using median imputation over other common methods (e.g., mean imputation, multiple imputation) would improve transparency. If a sensitivity analysis was conducted to assess whether the imputation method affected results, mentioning this would be valuable.

- The study is based on the BLSA cohort, which consists primarily of highly educated volunteers from a limited geographic region. This raises concerns about generalizability to more diverse populations, particularly individuals from different socioeconomic, racial, and educational backgrounds.

Suggested Improvement: The Discussion or Limitations section should briefly acknowledge this limitation and suggest future validation in more heterogeneous cohorts. Even a short statement recognizing the potential biases of volunteer-based longitudinal studies would enhance transparency.

- While the results clearly compare NN and LR, a concise summary statement highlighting the key takeaways—whether NN significantly outperformed LR or if the results were comparable—would help readers quickly grasp the implications.

Suggested Improvement: A small summary table comparing the strengths and weaknesses of both models in terms of accuracy, interpretability, and clinical applicability would make this section more accessible.

Specific Comments

1. Introduction

- Line 50-55: The claim that “gait speed is an essential predictor of overall health and well-being” is well-supported, but additional references on the impact of gait speed on cognitive decline would strengthen the argument.

- Line 72-75: The statement that NNs allow for capturing nonlinear relationships should be supported with examples from prior research in aging.

2. Methods

- Line 160-162: The handling of missing MMSE data (15%) is a potential limitation. Was a sensitivity analysis conducted to assess the impact of missing data on results?

- Line 231-234: The manuscript mentions using different solvers for logistic regression. Was feature selection or regularization applied to avoid overfitting?

3. Results

- Table 3: The dataset sizes after class balancing should be contextualized. How does this compare to the original dataset?

4. Discussion

- Line 357-363: The claim that NNs are advantageous for handling images and longitudinal data is valid but not demonstrated in this study. Consider clarifying that this is a potential future direction.

- Line 472-475: The discussion on modifiable risk factors (BMI, grip strength) is strong but could benefit from practical implications for clinical interventions.

5. Conclusion

- Line 607-612: The conclusion suggests that deep learning should be further explored, but it should acknowledge that logistic regression performed similarly, questioning the necessity of NN in this context.

**Do you want your identity to be public for this peer review?** For information about this choice, including consent withdrawal, please see our Privacy Policy

Reviewer #1: No

Reviewer #2: No

Reviewer #3: No

---

## [Author Response · Author response to Decision Letter 1]

24 Apr 2025

Reviewer #1:

1. I congratulate the authors for their work. Considering that the decrease in walking speed is an indicator of mortality for elderly individuals with chronic diseases, your work is very valuable.

a. We greatly appreciate Reviewer #1’s positive comments.

2. However, I have a few suggestions and questions. The intended use of deep learning and NN in elderly individuals with chronic diseases should be detailed in the introduction section.

a. We interpret this comment as suggesting that we note the eventual use of this in the clinical setting, which we agree is an excellent idea. We have provided potential applications and example use cases in a new paragraph in the introduction beginning, “The development of models…”

b. We have also further emphasized the applicability to a wide range of chronic diseases, with a new sentence and 3 new references in the introduction. The added sentence begins, “Gait speed is a metric of particular relevance for those with long-term, chronic conditions...”

3. The fact that the walking speed of the individuals was not measured with an objective equipment can be stated as a limitation.

a. We apologize for the evident lack of clarity on this point. We have edited the methods to emphasize that gait speed was measured using standard assessment methodologies based on usual current practice by trained observers, although we readily acknowledge that gait lab assessment would indeed be more precise. We have added two citations and a clarification to the methods section that states, “The SPPB, including its gait measurement component, is a standardized…”

b. We have now also included this as a minor limitation. It is unlikely that over the time frame of the timed walks, an improvement in measurement accuracy of ~1 second would result in any changes in classification in general and in our results in particular. However, acknowledging the Reviewer’s point, we have added a sentence to the discussion beginning, “Another potential limitation is that gait speed was measured by trained observers rather than by truly objective operator-independent methods.”

4. Only the right side was used for grip strength. Dominancy was not taken into concern?

a. We appreciate this consideration. We did have data from both hands for most participants; however, we evaluated their correlation and found significant collinearity, so we used only the right side. Indeed, we could have instead used “dominant hand”; this however would have forced us to use a somewhat smaller dataset as not every participant has their dominant hand noted. We have added a comment on the choice of only right side to the Data section 2.1.2 beginning with, “In fact, we elected to incorporate only the right-hand grip strength in the analysis…” We do acknowledge that a different choice could equally well have been made regarding incorporating left-sided grip strength.

5. Measurements of participants' walking speed over time may vary because of the individuals performing the measurements. It should be stated how this standardization was achieved. Was interrater reliability performed? ICC value?

a. We agree that these are important issues. First, we note that the BLSA protocols are long-established as high-quality assessments, and we made use of the generated data rather than re-collecting new data. Otherwise, interrater reliability and ICC are not included in the BLSA dataset. However, other studies which we have now cited have explored these or similar metrics. This resulted in our estimate of uncertainty of gait speed (0.06-0.11 m/s). Further exploration of the literature prompted by this comment indicated that ICC values for the timed walk tests range from 0.87 to 0.97. However, we were unable to find measurements of interrater reliability in the literature. We have now listed the ICC values in the Dataset Section 2.1.1 alongside the uncertainty values in conjunction with these additional citations. We have also, in line with this excellent point, incorporated these considerations into the limitations, beginning with, “Another potential limitation is that gait speed was measured by trained observers…”

6. In your study, age, grip strength and BMI were reported as major predictors similar to the results in the literature. LR and NN results are also similar. In this case, the superiority of NN or the reason for its preference is insufficient. This section needs to be supported by the authors.

a. We agree with this point and hope that we have emphasized that for this particular study, the performance of the NN and LR were similar. Of course, the additional flexibility afforded by the NN may prove to be crucial for other studies of gait and related physical outcomes. We have added a short paragraph in the Discussion Section 4.1.1 (discussing the optimized models’ performances) to remind readers of the advantages of the NN. In fact, we are currently completing another study in which a NN is used for gait prediction in which we evaluated use of input variables that are both tabular data and images. This is readily done with the NN formalism but cannot be accomplished with logistic regression in any conventional manner, since LR does not take images as input. The new paragraph begins, “While the performance difference between the NN and the LR was minimal in the present study…”

b. Note that to avoid further lengthening this already-lengthy manuscript, we kept this addition to the discussion rather short. More extensive arguments for the advantages of the NN are already included in both the introduction (“In addition to providing a natural means of developing implicit nonlinear models…”) and the conclusion (“The novel use of a NN for this purpose…”).

Reviewer #2:

1. This study addresses a highly relevant topic in the field, contributing valuable insights that enhance our understanding of gait and its determinants on aging. The authors have tackled an important question with a well-structured approach, and their findings have the potential to inform both clinical practice and future research. The thoroughness of the methodology and the depth of analysis further strengthen the study’s significance, making it a noteworthy addition to the existing literature. Nevertheless, I believe that implementing the suggested revisions will further enhance the quality and impact of the study, strengthening its contribution to the field.

a. We thank the reviewer for highlighting the potential significance of this work, and the detailed analysis we have provided.

2. Intro - The authors highlight the need for larger datasets in deep learning analyses and provide sample sizes from previous studies (108, 239, 746, 1901) as examples. However, no references are provided to support these figures. Including citations for these studies would enhance transparency and allow readers to verify the source of this information. (Line 81-86)

a. We apologize for this omission and thank the Reviewer for catching it. We have now provided the corresponding references.

3. Intro - The introduction would benefit from a clearer explanation of how the present study differs from previous research and which specific gap in the literature it aims to address. For instance, the statement 'Several attempts have been made using statistical models to predict gait speed changes from a narrow set of potential predictors' suggests prior work in this area. However, it would be helpful if the authors explicitly outlined how their study expands upon or improves these past efforts.

a. Thank you for this helpful comment. Although we did cite two papers with our initial comment, we have now further emphasized that as compared to prior work, the NN approach has the promise of greater flexibility and predictive power. We adjusted the sentence following the one noted by the reviewer to state: “These approaches, however, are restricted to the exploration of only linear relationships, while the complexity of human biochemistry and physiology suggests that their performance may be surpassed by that of models incorporating nonlinear effects and interactions.” We have also adjusted the paragraph flow to help connect this thought to the potential of the NN approach and indicate the possibility of expanding existing studies of nonlinear relationships to NN analysis.

b. Subsequently, we note in the MS: “Indeed, there remains a gap in the literature regarding the use of NNs for prediction of gait speed from clinical variables.” Using this as the topic sentence of a new paragraph emphasizes the Reviewer’s point and the specific gap we aim to fill.

c. We also call the reviewer’s attention to our Discussion Section 4.3.2 which fleshes out the models in the current literature and specifies in more detail the knowledge gap our work aims to fill.

4. Additionally, specifying the professional groups (health workers, physiotherapist, physician etc.) that could benefit from these findings would provide readers with a clearer understanding of the study’s relevance and impact.

a. We greatly appreciate this comment, which emphasizes the value in specifically addressing the important target audience of clinical practitioners. We have now specified the potential utility of this work to particular professional groups with a new paragraph in the introduction. The new paragraph begins, “The development of models that can accurately predict current and future gait speed decline would be of great clinical use in several contexts. For example...”

5. Methods - The primary aim of this study appears to focus on the relationship between gait speed, aging, and mortality. However, gait speed can be influenced by a wide range of factors, including orthopedic, neurological, and other medical conditions. Wouldn't it be more informative to stratify the analysis by different subgroups to account for these variations? Discussing the potential impact of such factors and whether a subgroup analysis could enhance the findings would strengthen the study's interpretation and applicability.

a. We greatly appreciate this comment. This represents a common trade-off in clinical studies between generalizability (using data from a large, heterogeneous, population) and accuracy (through limitation of confounding factors). Stratification by subgroup would, in principle, lead to greater accuracy within that subgroup, but would diminish the generalizability of the model to broad patient populations. In fact, gait speed is of great interest and importance precisely because, as the Reviewer points out, it integrates into one type of measurement a wide range of underlying pathologies. This makes it an excellent candidate measure for generalized datasets. We have now more specially highlighted this in the introduction with the addition of the sentence, “Indeed, one of the chief advantages of gait speed as a health metric is the fact that it is impacted by the integrity of a wide range of organ systems, including neurological status (both sensory and motor), cardiovascular health, orthopedic status, and pulmonary function.” Stratification would, therefore, result in study with a more limited and specific role for gait, in contrast to the present study in which gait is viewed as a broad-based integrated outcome. Of course, both approaches would have value, but for the present work, we elected to view gait speed as a final common indicator for multi-system and multi-organ function. In addition, especially in the context of NN analysis, further stratification would have a strongly negative impact on training capability by reducing dataset size. We have now incorporated this into the Limitations in a new paragraph that begins, “The BLSA dataset includes participants with a wide range of non-debilitating medical conditions.”

6. Methods - I acknowledge that biostatistics and methodological details can sometimes be complex to fully follow, so I apologize if I have missed anything. That being said, I would like to raise a few points regarding the chosen cut-points and the study population. The authors have established clinically relevant cut-points for gait speed (0.8 m/s and 1.0 m/s) based on prior literature. However, gait speed thresholds may vary across different populations, particularly in individuals with neurological conditions such as Parkinson’s disease or stroke. Were disease-specific variations in cut-points considered when applying these thresholds to the study population? If not, this could be a limitation, as the same cut-points may not be appropriate for all individuals.

a. We greatly appreciate this thoughtful comment. In fact, for this work, we were specifically interested in individuals without debilitating underlying conditions, in accordance with the BLSA participant population. This allowed us to gain insight into normative aging trajectories. Indeed, further work with targeted populations (e.g. prediction of gait speed trajectories for subjects with neurodegenerative disease) may be of substantial clinical value. We have added a clarifying statement to the description of the cut-points and how they are appropriate for the BLSA dataset: “…cut-points that have been shown to be clinically relevant for the normatively aging population represented by the BLSA.”

7. Methods - Additionally, the study utilized gait speed data from the BLSA cohort, where measurement intervals varied by age. Since older individuals had more frequent assessments, could this have biased the long-term predictive model? Were adjustments made to account for the potential overrepresentation of gait speed decline in older participants? A discussion on how these factors may have influenced the results would strengthen the study’s findings.

a. We indeed had a similar concern during model development. We found that there were far more measurements for the middle range of ages, with the potential therefore for greater accuracy, than the extremes, with the potential for decreased accuracy. However, with adequate training set size, accuracy even at the extremes would be expected to match that in the middle range. Therefore, we wished to determine whether the per-age training data was 1) adequate at the extremes and more than adequate in the middle range, or 2) inadequate at the extremes but (potentially) adequate in the middle range. Accordingly, we evaluated a per-age histogram of false predictions and compared it to the histogram of ages. We found, in fact, a remarkably similar structure, with the number of false predictions tracking the number of measurements. This is a strong indicator of lack of bias.

Figure 1. Histograms of the ages of participants at all gait speed measurements (blue) and the false predictions from the NN model (orange).

b. We also performed a sensitivity study in which measurements from the most over-represented age groups were excluded to an increasing extent. For the current gait speed prediction models tested, we found that the precision-recall area under the curve for the NN varied by less than 0.08 when removing up to 75% of the middle age range of values. This further supports the lack of bias due to overrepresentation of certain ages in the dataset.

c. We very much appreciate the reviewer bringing up this point and have now included a reference to these sensitivity studies in the Consideration of Bias Section 2.1.2. The new paragraph begins, “Lastly, we considered the possibility of bias from the uneven distribution of participant ages…” We have also now clarified the particular visit schedule dependence on participant age in Section 2.1.1.

8. Methods - It would be better for the readers if the authors explained a point about the measurement and use of grip strength in their analysis. In the table describing input variables, grip strength is reported in kilograms (Table 2). However, it is specifically labeled as "hand grip muscles right (kg)." Could you clarify whether only the right hand was measured and analyzed? If so, what was the rationale for not including the left hand or using an alternative approach such as the mean of both hands or the dominant hand? Additionally, considering that grip strength was identified as a significant factor in the Sobol index analysis and highlighted as a basis for therapeutic intervention, do you believe tha

---

## [Decision Letter · Decision Letter 1]

Prediction of future aging-related slow gait and its determinants with deep learning and logistic regression

PONE-D-24-56052R1

Dear Dr. Deatsch,

We’re pleased to inform you that your manuscript has been judged scientifically suitable for publication and will be formally accepted for publication once it meets all outstanding technical requirements.

Kind regards,

Esedullah Akaras

Academic Editor

PLOS ONE

Additional Editor Comments (optional):

Reviewers' comments:

Reviewer's Responses to Questions

**Comments to the Author**

Reviewer #2: All comments have been addressed

Reviewer #3: All comments have been addressed

2. Is the manuscript technically sound, and do the data support the conclusions?

Reviewer #2: Yes

Reviewer #3: Yes

3. Has the statistical analysis been performed appropriately and rigorously?

Reviewer #2: Yes

Reviewer #3: Yes

4. Have the authors made all data underlying the findings in their manuscript fully available?

Reviewer #2: No

Reviewer #3: No

5. Is the manuscript presented in an intelligible fashion and written in standard English?

Reviewer #2: Yes

Reviewer #3: Yes

Reviewer #2: I would like to sincerely thank the authors for their comprehensive and thoughtful responses to the reviewer comments. It is evident that considerable effort has been made to address the concerns raised during the initial review. The revised manuscript demonstrates substantial improvement, both in terms of clarity and scientific rigor. The additional explanations, methodological clarifications, and textual revisions have significantly strengthened the work’s contribution to the field. In particular, the enhancements to the Introduction and Limitations sections, as well as the added justifications regarding model choices and variable selection, are commendable. The manuscript is now more robust, transparent, and accessible to a broader clinical and academic audience. I appreciate the authors' diligence and responsiveness throughout the revision process.

Reviewer #3: All changes made by the authors are adequate and addressed the concerns of the reviewer. Therefore, I consider that the paper, in its current version, meets the necessary requirements to be published in Plos One.

**Do you want your identity to be public for this peer review?** For information about this choice, including consent withdrawal, please see our Privacy Policy

Reviewer #2: No

Reviewer #3: **Yes: ** Eduardo Carballeira

---

## [Editor Report · Acceptance letter]

PONE-D-24-56052R1

PLOS ONE

Dear Dr. Deatsch,

I'm pleased to inform you that your manuscript has been deemed suitable for publication in PLOS ONE. Congratulations! Your manuscript is now being handed over to our production team.

Kind regards,

on behalf of

Dr. Esedullah Akaras

Academic Editor

PLOS ONE